# Alcohol Consumption in Nursing Students after the COVID-19 Lockdown

**DOI:** 10.3390/healthcare11081185

**Published:** 2023-04-20

**Authors:** Carmen María Guerrero-Agenjo, Jaime López-Tendero, Ángel López-González, Isabel María Guisado-Requena, Jose Alberto Laredo-Aguilera, Juan Manuel Carmona-Torres, Jesús López-Torres-Hidalgo, Joseba Rabanales-Sotos

**Affiliations:** 1Castilla-La Mancha Health Service (Servido de Salud de Castilla-La Mancha/SESCAM), Universidad de Castilla-La Mancha/UCLM, 02071 Albacete, Spain; cmguerrero@sescam.jccm.es (C.M.G.-A.); jaimel@sescam.jccm.es (J.L.-T.); jesusl@sescam.org (J.L.-T.-H.); 2Department of Nursing, Physiotherapy and Occupational Therapy, Facultad de Enfermería, Universidad de Castilla-La Mancha, Campus Univesitario s/n, 02071 Albacete, Spain; isabelm.guisado@uclm.es (I.M.G.-R.); josealberto.laredo@uclm.es (J.A.L.-A.); juanmanuel.carmona@uclm.es (J.M.C.-T.); joseba.rabanales@uclm.es (J.R.-S.); 3Group of Preventive Activities in the University Health Sciences Setting (UCLM), 02071 Albacete, Spain; 4Grupo de Investigación Multidisplinar en Cuidados (IMCU) (UCLM), 45071 Toledo, Spain; 5Albacete Faculty of Medicine (UCLM), 02071 Albacete, Spain

**Keywords:** COVID-19, health, students, nursing, Spain

## Abstract

Excessive alcohol consumption is a major public health issue that can negatively affect behavior among university students. The objective of this study was to estimate the frequency of alcohol consumption in nursing students as well as to describe the pattern of alcohol consumption after COVID-19 lockdown. A descriptive, cross-sectional observational study was carried out, in which 1162 degree-level nursing students were evaluated. Sociodemographic characteristics, lifestyles and levels of physical activity were determined using the International Physical Activity Questionnaire: Short Form (IPAQ-SF), and alcohol consumption was determined using the ISCA (Systematized Alcohol Consumption Questionnaire) and AUDIT (Alcohol Use Disorders Inventory Test) questionnaires. According to the AUDIT questionnaire, 36.7% of the students met the criteria for excessive alcohol consumption (26.8% men vs. 39.9% women; *p* < 0.001). The prevalence of hazardous drinkers was found to be 10.2% (95% CI 5.6–11.7), with the difference between men and women being statistically significant. The IPAQ-SF questionnaire indicated that 26.1% of students were sedentary. No relationship was observed between alcohol consumption and the level of physical activity. The frequency of hazardous drinkers was significantly higher in women (OR: 2.2) and in smokers (OR: 4.2). In conclusion, approximately 10% of nursing students can be considered hazardous drinkers, with significant differences between the sexes. The percentage is higher in women and in smokers. Strategies should be created that encourage healthy lifestyles, emphasizing preventive activities against excessive alcohol consumption. Furthermore, given the differences in excessive alcohol consumption between men and women, it would be advisable to include the gender perspective in these activities.

## 1. Introduction

Excessive alcohol consumption is considered a major public health problem, as it is ranked among the top five risk factors for disease, disability and death worldwide. It was also identified as the second-most prevalent disease risk factor in Europe and the first-most in America and the Western Pacific countries [1]. Alcohol consumption is associated with neuropsychiatric diseases (depression, anxiety, alcohol dependence, etc.) [1], digestive system disorders (liver cirrhosis, pancreatitis, alcoholic hepatitis, etc.) [2], cardiovascular system disorders (arterial hypertension, atrial fibrillation, cardiovascular accidents, heart disease, etc.) [3] and even some cancers (mouth, larynx, esophagus, pancreas, etc.) [4]. In addition, alcohol abuse is often associated with smoking and the use of other drugs [1].

Globally, binge drinking causes approximately 3.3 million deaths each year, which is equivalent to 5.9% of all deaths. The morbidity attributable to this cause constitutes 5.1% of the global burden and carries a high socioeconomic cost. According to the World Health Organization (WHO), most alcohol-related problems appear in individuals who engage in risky or harmful consumption, with hazardous drinkers constituting a major concern in the field of preventive activities [1].

A differentiation must be made between hazardous drinkers and individuals who have developed alcohol dependence syndrome. The former is defined as the consumption of more than 280 g (93 oz) of alcohol per week for men and 168 g (56 oz) of alcohol per week for women, which is equivalent to 28 and 17 standard drinking units (SBU), respectively [5]. However, an individual with alcohol dependence syndrome presents a pattern of harmful consumption with alcohol and can be identified using the diagnostic criteria established in the DSM-V (Diagnostic and Statistical Manual of Mental Disorders) [6] or by the ICD-11 criteria in its section on substance use disorders (International Classification of Diseases and Other Health Problems) [7].

In general, excessive alcohol consumption is common among young people in industrialized countries both within and outside of Europe. In recent years, alcohol consumption in Spain has significantly decreased [8]; Spain moved from being the 6th highest consumer of alcohol worldwide in 2010 to the 29th highest in 2015 (WHO, 2018). Despite this, alcohol continues to be the most consumed psychoactive substance in Spanish people between 15 and 64 years of age, with the age of onset being between 13 and 16 years old and the prevalence of consumption increasing with age [9].

According to data from the National Plan on Drugs, the prevalence of hazardous drinkers in the general Spanish population fell from 6.5% in 2009 to 5.0% in 2015, with this decrease being greater in men than in women. This prevalence is higher in secondary and higher education students (15–24 years), reaching 5.3% in women and 3.9% in men [10].

In 2018, almost half of young people (46.5% of girls and 42.3% of boys) confessed to being drunk at some time in the last year compared to 21.8% in 2016. Additionally, the proportion of young people who had gotten drunk in the last month increased from 21.8% in 2016 to 32.3% in 2018 [10].

The proportion of hazardous drinkers is 37.1% in Spanish university students [11]. The results vary between 11.9 and 33.0% in medicine students [12] and between 17.9 and 27.2% in nursing students [13]. According to Rodriguez-Muñoz et al. [14], in Spain, 83.7% of nursing students consume alcohol, with women consuming more alcohol than men (85.5% vs. 75.0%). Of the total number of drinkers, 97.2% report that they consume a greater amount of alcohol during non-working days, with no significant differences between sexes or in terms of the place of consumption [14]. On the other hand, the age of onset is earlier when other toxic substances such as tobacco and marijuana are consumed, and alcohol consumption is more frequent among nursing students who consume psychoactive substances [15].

On 11 March 2020, the WHO declared the situation of a global pandemic caused by COVID-19. The Royal Decree (463/2020) of 14 March declared a state of alarm in Spain regarding the management of the health crisis caused by COVID-19 (2020 Royal Decree: Real Decreto Boletin Oficial del Estado 463/2020) [16]. This involved a series of measures, including the cessation of face-to-face teaching (including teaching in universities), until the end of the 2019–2020 academic year, during which students returned to their families’ homes, a factor associated with lower alcohol consumption [17].

At the beginning of the academic year (September 2020), most university students had returned to face-to-face teaching, although with restrictions. Some of these were directly associated with changes in the pattern of alcohol consumption [18], such as limiting the number of attendees at social gatherings, the closure of nightlife and sports venues and the implementation of a curfew. On the other hand, the current COVID-19 pandemic is causing health problems in young people such as stress, worry, fear, anxiety and depression [19,20], risk factors that can trigger changes in alcohol consumption.

The risk of alcohol consumption is associated with various risk factors for university students themselves, such as intoxication, tobacco consumption, sedentary lifestyle, premature cardiovascular disease and substance dependence, as well as risks to third parties, such as accidents, violence, risky sexual behaviors, etc. [21]. In addition, a lack of physical exercise and a sedentary lifestyle were risk factors whose prevalence increased in Spanish university students during the COVID-19 lockdown, although lockdown was reported as a protective factor against previous toxic habits such as excessive drinking and smoking [22], as is the case for university students in Portugal [23].

It is therefore important to study alcohol consumption patterns during the “new normal” period following the COVID-19 pandemic.

The aim of this study was to estimate the frequency of alcohol consumption among nursing students after the COVID-19 lockdown. In addition, it had the secondary objective of describing the most common patterns of behavior in relation to heavy drinking after COVID-19.

## 2. Materials and Methods

### 2.1. Design

A descriptive cross-sectional study was carried out.

### 2.2. Sample/Participants

The study was carried out on undergraduate nursing students enrolled at the University of Castilla-La Mancha (Spain) during the 2021–2022 academic year. The inclusion criteria were as follows: student enrolled in the nursing degree program at one of the nursing faculties of the University of Castilla-La Mancha and willing to participate in the study.

The sample size was calculated using the GRANMO program (v 7.12 April 2012) for population estimation (proportions). For a confidence level of 95% and an accuracy of ±2% units, 14% of students were expected to be hazardous drinkers, with a predicted replacement rate of 10%; a random sample of 693 individuals was determined to be sufficient.

A total of 1500 students were invited via e-mail to participate in the study. Before participating in the study, all students had to read information that informed them in detail about the purpose of the study. In addition, they had to give their informed consent. A total of 1162 students responded; thus, the response rate was 77.4%.

### 2.3. Data Collection

A pre-coded data collection form was designed. The questionnaire was anonymously self-administered. The forms were completed between September and October 2021.

The study variables were as follows: sociodemographic characteristics (age, sex, social class based on occupation of students’ parents (the main earners of a household) (CSO-SEE12) [24] and rural or urban environment); lifestyle (toxic health habits such as use of tobacco or other substances (cannabis, cocaine, etc.) (according to the WHO, smokers are those who have smoked any number of cigarettes daily during the last month, including one cigarette per day); level of physical activity using the International Physical Activity Questionnaire: Short Form (IPAQ-SF) [25] (classifying students as sedentary, partially active or active); assessment of hazardous alcohol consumption and alcohol dependence using the Alcohol Use Disorders Inventory Test (AUDIT) (Spanish version) [26]; quantification of alcohol consumption using the Interrogatorio Sistematizado de Consumo Alcohólico (ISCA) [27].

### 2.4. Ethical Considerations

This study was approved by the Ethics and Clinical Research Committee of Albacete (Spain) with protocol number (2020/07/072). The research complied with the principles of the Declaration of Helsinki. To preserve the confidentiality of the participants, the questionnaire data were entered into a database and exclusively identified using a numerical code both on the data collection form and in the computerized database (2018 Personal Data Protection Act: Ley de Protección de Datos Personales y Garantía de los Derechos Digitales, 3/2018) [28].

### 2.5. Data-Analysis

The data were analyzed using IBM SPSS Statistic v.24.0 (IBM Corp. Armonk, NY, USA), which was licensed by the Universidad de Castilla La-Mancha. Participant characteristics were analyzed using counts (n) and proportions (%) for qualitative variables and measures of central tendency (means (m) and standard deviation (SD)) for quantitative variables. The chi-square test (χ2) was used to compare proportions between groups, and the Mantel–Haenszel test was used for linear trends in the case of ordinal variables. Likewise, Student’s *t*-test was used to compare means in two groups. Correlation analysis was performed using Spearman’s rho for linear associations between AUDIT and the number of SBU. Finally, to independently evaluate the factors associated with the presence of hazardous drinker criteria, as a response variable, a logistic regression model was implemented using the Wald statistic, in which variables with *p* ≥ 0.15 were eliminated.

The AUDIT is an international self-administered questionnaire used to detect non-serious alcohol-related problems, and it was validated for use in Spain [26]. It consists of 10 questions to detect mild and moderate alcohol-related problems, focusing on hazardous drinking or alcohol abuse. The reliability (Cronbach’s alpha) is 0.932. In addition, the ISCA is a validated questionnaire consisting of three questions to measure the amount and frequency of alcohol consumption (number of beers, wines and spirits), differentiating between working days and weekends/holidays, and allows for the quantification of the weekly intakes considered hazardous by the WHO (28 SBU for men and 17 for women) [27]. The sensitivity of ISCA in men is 70–81%, and their specificity is 82–99%. In women, their sensitivity is 46–100%, and their specificity is 97–100%. Finally, the IPAQ-SF is a self-administered questionnaire, also validated in Spain [25], that measures the frequency (in days per week), duration (times per day) and intensity (moderate or vigorous) of activities over the previous 7 days. Its reliability is 0.7.

## 3. Results

In total, 1162 nursing students were analyzed; their mean age was 21.2 years (SD 3.5), and the majority were female (75.6%). The sociodemographic and lifestyle characteristics of these students are shown in Table 1.

### 3.1. Prevalence of Hazardous Drinkers

The responses obtained in the AUDIT Questionnaire are shown in Table 2, where it is observed that 20.0% of the students consumed alcohol at least two or three times a week, 35% consumed three or four drinks during a typical drinking day, and 21.6% consumed five or more drinks. A total of 33.4% of students claimed to have consumed six or more drinks on a single occasion less than once per month, whereas 21.1% did so at least once per week. Considering the cut-off points used for men and women, the rate of harmful consumption of alcohol reached 26.8% and 39.9%, respectively, with the differences being statistically significant (*p* > 0.001). In the sample as a whole, the prevalence was 36.7%.

Table 3 shows the mean number of SBU according to the types of drinks consumed by men and women based on the responses to the ISCA questionnaire. In the total sample, the mean consumption was 8.88 (SD: 9.22), being higher in men than in women (9.46 ± 10.4 SD vs. 8.69 ± 8.8 SD; Student’s *t*-test; *p* = 0.077) but not with statistically significant differences. According to the results of the questionnaire, 69.5% of the students stated that they changed their consumption habits during the weekend. Considering risk drinking limits in men and women, the prevalence values of hazardous drinkers were 5.6% and 11.7%, respectively, the difference being statistically significant: *p* = 0.003. In the whole sample, the prevalence was 10.2% (CI: 95% 5.6–11.7). A high correlation was observed between the score obtained in the AUDIT questionnaire and the number of SBUs, which was statistically significant (Spearman’s Rho: 0.620; *p* < 0.001).

### 3.2. Variables Associated with Hazardous Alcohol Consumption

The proportion of hazardous drinkers was significantly higher in women (11.7% vs. 5.6%; likelihood ratio Chi-square test, *p* = 0.003) and in smokers (25.1% vs. 7.5%; likelihood ratio Chi-square test, *p* < 0.001). No statistically significant differences (*p* = 0.295) were observed between the proportion of irrigation drinkers and social class (Figure 1). In addition, a non-statistically significant linear trend (Mantel–Haenszel test for linear trends, *p* = 0.056) was observed between a higher proportion of hazardous drinkers and a lower number of inhabitants in the population of origin (15.4% in populations of less than 10,000 inhabitants, 12.2 % in populations from 10,001 to 40,000 and 9.1% in populations over 40,000). Regarding age, the mean value was lower in students with hazardous drinker criteria (20.6 years ± 2.8 SD vs. 21.2 years ± 3.6 SD; *p* = 0.077).

Using logistic regression (Table 4), it was found that the frequency of hazardous drinkers was significantly higher in women (OR: 2.2) and in smokers (OR: 4.2).

## 4. Discussion

Nursing students assessed their alcohol consumption (through the AUDIT and ISCA questionnaires) after the COVID-19 pandemic, determining whether their alcohol-related habits could potentially be classed as hazardous or harmful drinking.

As supported by the results obtained in our study, several authors describe alcohol consumption as a cultural element that is reinforced in a university environment [29]. Most young university students say they consume alcohol during their leisure time and consider it a social disinhibitor [30]. In the university setting, the pattern of consumption is strongly linked to the phenomenon of the “botellón” (street drinking), characterized by intensive “bingeing” during the weekends [31], which is also shown in the results of our study. In the results of the AUDIT questionnaire, we observed that a high proportion of students (36.7%) presented a positive result through the cut-off points used; however, the proportion of hazardous drinkers according to the ISCA questionnaire was (10.2%), resulting in a clearly lower rate in addition to observing a statistically significant difference between women and men when considering the limits of specific risks. As described in other studies in the university setting, women consume more harmful substances than men [14], although there is controversy in the scientific literature in this regard. Several studies carried out on university students (and some on nursing students) in Europe, Asia, South America and Australia conclude that male students are more likely to drink excessively [17,32,33,34], although some of them point to a narrowing of the gender gap in binge drinking [18,35]. Previous research has associated this phenomenon with differentiated social norms between genders and cultural settings [34,35].

It should be noted that consumption is higher in those students who live outside the family nucleus, which coincides with the results of other studies, suggesting that young people who reside in the family home during the university period are less prone to the intensive consumption of alcohol than those who live on campus or away from home [32,34].

In our sample, the percentage of smokers was very low (15%) and comparable to that of other health science university students [22], with no difference being observed between men and women. It should be noted that almost one quarter of students who met the criteria for hazardous drinkers were smokers; this was previously described in studies carried out in university students [13,17].

University students, including those from health science, show little adherence to physical activity [20,36]. Some authors have described that during the COVID-19 lockdown, young people, mainly women, increased their level of physical activity [20]. Men performed more physical activity than women, a fact also described in the university setting [25]. In our study, no significant relationship was found between the intensity of physical activity and alcohol consumption, a fact also previously described by other authors [35]. However, this issue is controversial because some studies found an association between higher levels of physical activity and higher alcohol consumption [17], whereas others point out that physical activity could be a protective factor against excessive alcohol consumption [25,35].

There was a higher proportion of hazardous drinkers among students who come from a population with fewer inhabitants, contrary to that described by Zadarko et al. in a 2018 study of Polish university students [34]. The same occurred with younger university students, a finding supported by a study of Australian university students [33]. On the contrary, another study carried out among nursing students in Spain concluded that students at more advanced stages of their courses drink more alcohol than those in their first year [25].

If we compare the proportion of hazardous drinkers among the students in our sample according to their social class, we do not find statistically significant differences. In the consulted bibliography, there is no worldwide consensus on the relationship between social class and the proportion of hazardous drinkers, as some authors associate a higher social class with higher risk consumption [13,34], and other authors consider lower social class as a protective factor against alcohol consumption [32].

### Limitations

This study has some limitations. First, the data are self-reported. Because the data were collected in class, it could be that people who did not attend class, and therefore did not participate in the study, had different patterns of alcohol consumption and/or different sociodemographic variables. The study design did not allow us to establish causality between the different variables. Additionally, women having a higher risk for alcohol abuse could be due to self-reporting. The results may have been affected because women were more honest in answering the questionnaire, and because men may underestimate their consumption. Future studies along these lines would be useful to establish solid conclusions.

However, the greatest strength of this study is that it uses a representative sample of university nursing students in Spain and is the first study in this county to analyze alcohol consumption in nursing students after the COVID-19 pandemic, a period in which many changes in students’ lifestyles may have affected their alcohol consumption.

## 5. Conclusions

In conclusion, the results show that about 10% of nursing students can be considered hazardous drinkers, with significant differences between sexes. The proportion of hazardous drinkers is higher in women and smokers.

In the future, further studies should be conducted to learn about modifiable risk factors that can prevent the consequences of alcohol abuse at these ages. More efforts should be made to create strategies that motivate students towards a healthier lifestyle with an emphasis on preventive activities against excessive alcohol consumption in the university environment. These activities would in turn introduce them to the field of preventive measures against alcohol, promoting their ability to detect hazardous alcohol consumption as future health professionals. Therefore, it is necessary to promote healthy activities among nursing students (decreasing their alcohol consumption). These students will soon be health professionals, and they should be aware of the harmful effects of excessive alcohol consumption and be able to help those who consume excessive amounts of alcohol.

Given that there are significant differences between the excessive consumption of alcohol between men and women, and due to the controversy described in the literature, it would be advisable to include gender perspectives in these activities.

## Figures and Tables

**Figure 1 healthcare-11-01185-f001:**
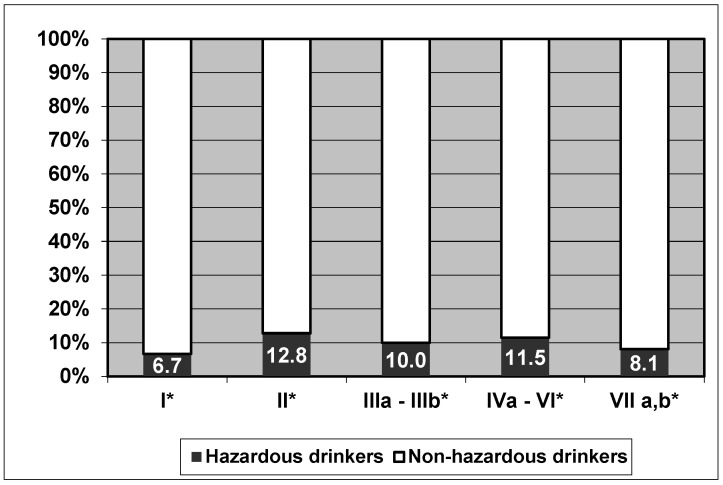
Proportion of hazardous drinkers by social class. I*: Higher-grade professionals, administrators and officials; managers in large industrial establishments; large proprietors. II*: Lower-grade professionals, administrators and officials; higher-grade technicians; managers in small industrial establishments; supervisors of non-manual employees. IIIa*: Routine non-manual employees, higher grade (administration and commerce). IIIb*: Routine non-manual employees, lower grade (sales and services). IVa: Small proprietors, artisans, etc. with employees. IVb: Small proprietors, artisans, etc. without employees. IVc: Farmers and smallholders; other self-employed workers in primary production. V: Lower-grade technicians; supervisors of manual workers. VI*: Skilled manual workers. VIIa*: Semi-skilled and unskilled manual workers (not in agriculture, etc.). VIIb*: Agricultural and other workers in primary production.

**Table 1 healthcare-11-01185-t001:** Characteristics of study participants.

Characteristics	Men	Women	Total
	n (%)	n (%)	n (%)
**Age (years)**			
18–19	101 (35.9)	275 (31.3)	377 (32.4)
20–21	91 (32.0)	345 (39.3)	436 (37.5)
22–23	51 (18.0)	139 (15.8)	190 (16.4)
24 or over	40 (14.1)	119 (13.6)	159 (13.7)
**Academic year**			
First	75 (26.4)	225 (25.6)	300 (25.8)
Second	97 (34.2)	204 (23.2)	301 (25.9)
Third	65 (22.9)	257 (29.3)	322 (27.7)
Fourth	47 (16.5)	192 (21.9)	239 (20.6)
**Nursing faculty**			
Albacete	90 (31.7)	313 (35.6)	403 (34.7)
Ciudad Real	86 (30.3)	226 (25.7)	312 (26.9)
Cuenca	72 (25.4)	172 (19.6)	244 (21.0)
Toledo	26 (9.2)	87 (9.9)	113 (9.7)
Talavera de la Reina	10 (3.5)	80 (9.1)	90 (7.7)
**Population of origin (inhabitants)**			
Under 10,000	21 (7.4)	117 (13.3)	138 (11.9)
10,000 to 40,000	21 (7.4)	103 (11.7)	124 (10.7)
Over 40,000	242 (85.2)	658 (74.9)	900 (77.5)
**Living situation**			
Living alone	10 (3.5)	26 (3.0)	36 (3.1)
Living with partner or married, with/without children	17 (6.0)	49 (5.0)	66 (5.7)
Living with parents, siblings, etc.	149 (52.5)	466 (53.1)	615 (52.9)
Living in shared accommodation	82 (28.9)	268 (30.5)	350 (30.1)
Living in a residence	26 (9.2)	69 (7.9)	95 (8.2)
**Smoker**			
Yes	41 (14.4)	140 (15.9)	181 (15.6)
No	243 (85.6)	738 (84.1)	981 (84.4)
**Physical exercise**			
Sedentary	57 (20.1)	246 (28.0)	303 (26.1)
Partially active	77 (27.1)	333 (37.9)	410 (35.3)
Active	150 (52.8)	299 (34.1)	449 (38.6)
**Social class ***			
I	42 (16.3)	92 (11.1)	134 (12.3)
II	48 (18.6)	132 (15.9)	180 (16.5)
III (a,b)	40 (15.5)	190 (22.9)	230 (21.1)
IV (a,b,c)-VI	54 (20.9)	200 (24.1)	254 (23.3)
VII (a,b)	72 (27.9)	213 (25.7)	285 (26.2)
Not shown	2 (0.8)	3 (0.4)	5 (0.5)

* I: Higher-grade professionals, administrators and officials; managers in large industrial establishments; large proprietors. II: Lower-grade professionals, administrators and officials, higher-grade technicians; managers in small industrial establishments; supervisors of non-manual employees. IIIa: Routine non-manual employees, higher grade (administration and commerce). IIIb: Routine non-manual employees, lower grade (sales and services). IVa: Small proprietors, artisans, etc. with employees. IVb: Small proprietors, artisans, etc., without employees. IVc: Farmers and smallholders; other self-employed workers in primary production. V: Lower-grade technicians; supervisors of manual workers. VI: Skilled manual workers. VIIa: Semi-skilled and unskilled manual workers (not in agriculture, etc.). VIIb: Agricultural and other workers in primary production.

**Table 2 healthcare-11-01185-t002:** Distribution of answers to the AUDIT * questionnaire.

Questions	Men n (%)	Women n (%)	Total n (%)	*p*
**How often do you have a “drink” containing alcohol?**				
Never	33 (2.8)	80 (6.9)	113 (9.7)	**0.005**
Once or less per month	60 (5.2)	205 (17.6)	265 (22.8)	
2–4 times per month	115 (9.9)	437 (37.6)	552 (47.5)	
2–3 times per week	65 (5.6)	138 (11.9)	203 (17.5)	
4 or more times per week	11 (0.9)	18 (1.5)	29 (2.5)	
**How many “alcoholic drinks” do you have on a typical day when you have been drinking?**				
0–2	138 (11.9)	364 (31.3)	502 (43.2)	**0.008**
3–4	78 (6.7)	307 (28.3)	407 (35.0)	
5–6	48 (4.1)	152 (13.1)	200 (17.24)	
7–9	13 (1.1)	23 (2.0)	36 (3.1)	
10 or more	7 (0.6)	10 (0.9)	17 (1.5)	
**How often do you have six “drinks” or more on a single occasion?**				
Never	115 (9.9)	425 (36.6)	540 (46.5)	**0.013**
Less than once per month	89 (7.7)	288 (24.8)	377 (32.4)	
Monthly	52 (4.5)	104 (9.0)	156 (13.4)	
Weekly	28 (2.4)	60 (5.2)	88 (7.6)	
Daily or almost daily	0 (0.0)	1 (0.1)	1 (0.1)	
**How often in the last year,** have **you felt unable to stop drinking once you have begun?**				
Never	237 (20.4)	770 (66.3)	1007 (86.7)	**0.048**
Less than once per month	34 (2.9)	89 (7.7)	123 (10.6)	
Monthly	11 (0.9)	14 (1.2)	25 (2.2)	
Weekly	1 (0.1)	5 (0.4)	6 (0.5)	
Daily or almost daily	1 (0.1)	0 (0.0)	1 (0.1)	
**How often in the last year, have you been unable to do what you were normally expected to do due to drink?**				
Never	227 (19.5)	712 (61.3)	939 (80.8)	0.511
Less than once per month	50 (4.3)	149 (12.8)	199 (17.1)	
Monthly	7 (0.6)	12 (1.0)	19 (1.6)	
Weekly	0 (0.0)	4 (0.3)	4 (0.3)	
Daily or almost daily	0 (0.0)	1 (0.1)	1 (0.1)	
**During the last year, how often have you needed a shot of alcohol in the morning, in order to be able to “get going” after a night of heavy drinking?**				
Never	277 (23.8)	870 (74.9)	1147 (98.7)	0.130
Less than once per month	6 (0.5)	7 (0.6)	13 (1.1)	
Monthly	1 (0.1)	1 (0.1)	2 (0.2)	
Weekly	0 (0.0)	0 (0.0)	0 (0.0)	
Daily or almost daily	0 (0.0)	0 (0.0)	0 (0.0)	
**During the last year, how often have you felt guilt or remorse after having been drinking?**				
Never	158 (13.6)	508 (43.7)	666 (57.3)	0.194
Less than once per month	101 (8.7)	313 (26.9)	414 (35.6)	
Monthly	18 (1.5)	48 (4.1)	66 (5.7)	
Weekly	7 (0.6)	7 (0.6)	14 (1.2)	
Daily or almost daily	0 (0.0)	2 (0.2)	2 (0.21)	
**During the last year, how often have you been unable to recall what happened the previous night due to having been drinking?**				
Never	170 (14.6)	554 (47.7)	724 (62.3)	0.054
Less than once per month	102 (8.8)	263 (22.6)	365 (31.4)	
Monthly	9 (0.8)	58 (5.0)	67 (5.8)	
Weekly	2 (0.2)	2 (0.2)	4 (0.3)	
Daily or almost daily	1 (0.1)	1 (0.1)	2 (0.2)	
**Have you or any other person suffered any harm as a consequence of the fact that you had been drinking?**				
No	242 (20.8)	781 (67.2)	1023 (88.0)	0.141
Yes, but not in the last year	32 (2.8)	66 (5.7)	98 (8.4)	
Yes, in the last year	10 (0.9)	31 (2.7)	41 (3.5)	
**Has any relative, friend, doctor or health professional expressed concern at your drinking or suggested that you stop drinking?**				
No	254 (21.9)	851 (73.2)	1105 (95.1)	**<0.001**
Yes, but not in the last year	9 (0.8)	13 (1.1)	22 (1.9)	
Yes, in the last year	21 (1.8)	14 (1.2)	35 (3.0)	

* AUDIT: Alcohol Use Disorders Inventory Test. In bold type: *p* ≤ 0.05.

**Table 3 healthcare-11-01185-t003:** Mean number of SBU per week among men and women, according to type of alcohol beverage (Student’s *t*-test).

Type of Alcoholic Beverage	Mean Number of SBU * (95% CI)			*p*
	Men	Women	Total	
Beer	3.3 (2.8–3.8)	2.7 (2.4–2.9)	2.8 (2.6–3.1)	0.026
Wine	0.8 (0.6–1.1)	0.9 (0.8–1.0)	0.9 (0.8–1.0)	0.428
Spirits	2.7 (2.3–3.6)	2.6 (2.4–2.8)	2.6 (2.4–2.8)	0.594
All beverages	9.5 (8.3–10.7)	8.7 (8.1–9.3)	8.9 (8.4–9.4)	0.285

* SBU: standard drink units. In bold type: *p* ≤ 0.05.

**Table 4 healthcare-11-01185-t004:** Variables shown via multiple logistic regression to be associated with the presence of hazardous drinker criteria.

Variables	B	Wald	*p*	OR (95% CI)
Women vs. men	0.798	7.976	0.045	2.221 (1.277–3.865)
Smokers vs. non-smokers	1.429	45.306	<0.001	4.175 (2.754–6.330)

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
