# Peer review of "Alcohol Consumption in Nursing Students after the COVID-19 Lockdown"

_healthcare, 2023, doi:10.3390/healthcare11081185_

Round 1
Reviewer 1 Report
Thank you for submitting this article. This piece on alcohol use among nursing students during covid was very interesting.
Literature Review
In your literature review, I would add information on the health impacts on episodic or binge drinking has. I would also include information on risk factors for college students and binge/ episodic drinking (you include covid, but compare how those risk factors have changed before and after covid).
Below are the details of my feedback.
Line 25- you wrote that alcohol consumption was measured among Health Sciences students, but your title states nursing students. On Line 27, you stated Nursing degree students. I would make this clearer in the abstract if the nursing was taken from the overall health sciences students.
Line 59- the author uses grams which I assume is the preferred metric in their geographic region. Some readers may used ounces to measure standardized drinks. I would include both measurements if possible.
Methods and Materials
For the table... we use n(%), not no. (%).
I am also unsure what social class means? Are these students that also work? Please clarify in your instrumentation what these demographic variables mean.
Discussion
Your discussion is introducing new sources when it should be using the sources in your literature review to discuss the results... i.e. no new information should be introduced in your literature review.
Also, the phenomenon of having women being higher risk for alcohol abuse could be due to self reporting. Women were more honest and men underestimate their consumption (possible). Also, why were other genders not measured or sexual orientation? This could impact the outcomes.
Author Response
Dear reviewer,
We would like to thank you for giving us the opportunity to revise and improve our manuscript; we also thank the reviewer for yours thoughtful and constructive comments. We have considered all of the suggestions and have incorporated them into the revised manuscript, and as a result, we believe our manuscript is stronger. Responses to your comments are written in bold. We have highlighted the changes in the manuscript in yellow.

Reviewer 2 Report
Dear authors, please see the attached word file.

Author Response

(The authors gave the same response as above.)

Round 2
Reviewer 2 Report
The manuscript looks better. However, there are some comments, especially in the “Methodology section” that still need to be addressed to be changed. Please see the attached word file.

Author Response
We would like to thank you for giving us the opportunity to review again and improve our manuscript.
We have considered all suggestions and incorporated them into the revised manuscript, and as a result, we believe our manuscript is stronger. Responses to his comments are written in bold type. We have highlighted in yellow the changes made to the manuscript.

Round 3
Reviewer 2 Report
Dear authors, manuscript